# Simultaneous Photodiagnosis and Photodynamic Treatment of Metastatic Melanoma

**DOI:** 10.3390/molecules24173153

**Published:** 2019-08-29

**Authors:** Channay Naidoo, Cherie Ann Kruger, Heidi Abrahamse

**Affiliations:** Laser Research Centre, Faculty of Health Sciences, University of Johannesburg, Johannesburg 2028, South Africa

**Keywords:** metastatic melanoma (MM), photodynamic diagnosis (PDD), photodynamic therapy (PDT), photosensitizer (PS), nanoparticles (NP), 5-Aminolevulinic acid (5-ALA), hypericin, phthalocyanines

## Abstract

Metastatic melanoma (MM) has a poor prognosis and is attributed to late diagnoses only when metastases has already occurred. Thus, early diagnosis is crucial to improve its overall treatment efficacy. The standard diagnostic tools for MM are incisional biopsies and/or fine needle aspiration biopsies, while standard treatments involve surgery, chemotherapy, or irradiation therapy. The combination of photodynamic diagnosis (PDD) and therapy (PDT) utilizes a photosensitizer (PS) that, when excited by light of a low wavelength, can be used for fluorescent non-destructive diagnosis. However, when the same PS is activated at a higher wavelength of light, it can be cytotoxic and induce tumor destruction. This paper focuses on PS drugs that have been used for PDD as well as PDT treatment of MM. Furthermore, it emphasizes the need for continued investigation into enhanced PS delivery via active biomarkers and passive nanoparticle systems. This should improve PS drug absorption in MM cells and increase effectiveness of combinative photodynamic methods for the enhanced diagnosis and treatment of MM can become a reality.

## 1. Introduction

Skin cancer is classified as the irregular production and mutation of epithelial cells causing the malignant growth of the cells within the various layers of the skin allowing tumor formation [1]. According to the World Health Organization (WHO), for every three patients diagnosed with cancer, one of these patients will have skin cancer [2]. Moreover, due to the extended exposure of ultra-violet radiation and aging, the incidence rate of skin cancer is rising [3]. Skin cancer can be identified as either a non-melanoma, which consists of basal cell carcinoma and squamous cell carcinomas, or melanoma [1]. 

Metastatic melanoma (MM) is the sixth most diagnosed cancer globally; within the United States (US), there has been an annual 54% increase in cases being diagnosed between 2009 and 2019 [4,5]. Metastasis occurs when cancer cells advance to a secondary location by one of three ways: tumor direct growth into neighboring tissue; movement through the bloodstream via the circulatory system; and lastly, by traveling within the lymphatic system and localizing in regional lymph nodes [1]. Thus, MM is the deadliest form of skin cancer, due to this metastasis and once it metastasizes the identification of the primary and/or secondary location is difficult [1,6]. Therefore, early diagnosis of MM is essential, as diagnosing it at a later stage could lead to a poor prognosis, with limited treatment options, leading to a potentially fatal outcome [7]. Standard treatment for MM includes surgery, radiation therapy, biological therapy, and chemotherapy [8]. Recent advances have been made with immunotherapy and targeted therapy, for example checkpoint blockades which include the use of anti-cytotoxic T-lymphocyte antigens (CTLA-4), Programmed Death 1 (PD-1), and Programmed Death Ligand 1 or 2 (PD-L1/ 2, BRAF and MEK inhibitors) [8]. However, this type of treatment is not useful for every patient, as it is often affected by resistance factors and immune-related side effects [8].

Mostly, the primary detection of MM is performed by a dermatological examination of one’s skin in order to identify if new moles/nevi have developed or if existing moles/nevi have changed in size, shape, color, and/or border appearance [9]. Once identified, tumors can then be subjected to various imaging tests such as computed tomography (CT) scans and Positron emission tomography–computed tomography (PET-CT) scans for confirmatory identification [10]. Additionally, if tumors are identified, there are other standard diagnostic procedures, which are conducted to determine the stage and final diagnosis of MM. However, these standard diagnostic modalities have drawbacks and so there is an ongoing need to investigate novel and alternative diagnostic applications for MM.

The gold standard procedure for MM diagnosis is a biopsy. There are multiple types of biopsy’s available, but the two most common types applied within the diagnosis of MM is an incisional or fine needle aspiration biopsy [11]. If the melanoma cancer has not yet metastasized and remains superficially located on the skin, excisional biopsies are used, whereby the entire tumor mass in one small lesion can be removed [12]. However, incisional biopsies, for MM diagnosis often include shave and punch biopsies, to yield sufficient information for cancer progression and staging with an 88% accuracy in relation to Breslow’s depth [13]. However, this form of diagnosis has drawbacks due to its invasive nature and is limiting, as it does not allow for the identification of the primary or secondary origin of the tumor [11]. Fine needle aspiration biopsy is a fast, cost effective, and minimally invasive procedure, which involves the insertion of a hollow needle directly into the tumor mass to remove fluid, cells, and/or tissue [14]. This procedure has been proven to be extremely sensitive for MM detection. However, this procedure is also limiting due to the fact that it simply cannot determine the original source of MM and so effectivity of overall treatment is reduced and there is a high risk if MM reoccurrence due to metastasis [14]. Sentinel lymph node biopsy assesses regional lymph nodes in order to determine if the lymphatic system is responsible for MM metastasis [15]. 

Haematological and biochemical assays are not routinely used to diagnosis MM. However, they are used to monitor the effectiveness of a treatment that is administered. A blood test is used to determine the levels of a full blood count, vitamin D, or Lactate dehydrogenase (LDH). If there is an increase in white blood cells noted within a full blood count, this usually is an indication that the cancer treatment is not working [16]. A study by Timerman et al. (2017) noted that vitamin D deficiencies are usually linked to a low mortality rate of MM patients [17]. Lastly, elevated levels of LDH is usually an indicator for melanoma metastases [18]. 

Recently research has focused on developing alternative diagnostic tools for MM detection, due to the above-mentioned limitations (noted within standard methods), especially in relation to early detection. One such alternative method is known as mutational testing. Within this testing model BRAF (serine/threonine-protein kinase B-Raf) and NRAS (neuroblastoma RAS viral oncogene homolog) mutations are sequentially investigated for diagnostic purposes, as 70% of melanomas detect BRAF and NRAS as common driver mutations [19]. Additionally, mutational testing can be performed with miRNA, as miR-221/22 has the potential for more accurate MM diagnosis or matrix-assisted laser desorption ionization (MALDI) mass spectrometry to identify exosomal fingerprints to detect cancer biomarkers [20,21]. However, both techniques are still under investigation for standardized accuracy [20,21]. Lastly, overexpressed MM biomarkers identified in patients’ blood, sometimes has the ability to detect melanoma before it metastasizes, however blood sample analysis for diagnosis is mostly performed at late stages of MM and so are not useful within early stage diagnosis, thus it is mostly applied to assess a patients prognosis during treatment [11].

The development of imaging techniques for MM has become more desirable; since they are more reliable, swift, and feasible, as well as completely non-invasive and so pain-free [22].

Anatomic imaging uses rudimental principals for the identification of tumor masses within the body, as the premise of most anatomic imaging is the phenotypic change of tissues or organs within the body [22,23]. Current anatomical imaging used for MM diagnosis is high-frequency ultrasound (HFUS), confocal microscopy (CM), terahertz pulsed imaging (TPI), fluorescence remission sensoring, magnetic resonance imaging (MRI), optical coherence tomography (OCT), photoacoustic microscopy, and Raman spectroscopy [22]. However, the challenges that often accompany this type of imaging technique is that cancer cells may go undetected if no tumor mass is formed or if the tumor mass is extremely small [23]. Furthermore, tumors that are often identified using this type of imaging technique may either be benign or malignant. However, after the surgical removal of the detected tumor mass, it becomes difficult post-operative to detect if a recurrent tumor is present [23]. Additionally, anatomic imaging becomes rather limiting in relation to tissue depth penetration, tends to have a low resolution and sensitivity, as well as is expensive and requires extensive training of diagnostic personal [22]. Molecular imaging is defined as the visualization, classification and measurement of biological processes at a cellular level utilizing an imaging agent [22]. Current molecular imaging used include optical bioluminescence, molecular MRI (mMRI), optical fluorescence, magnetic resonance spectroscopy (MRS), positron emission tomography (PET), targeted ultrasound, and single photon emission computed tomography (SPECT) [22]. The disadvantages of molecular imaging include limitations in imaging that are dependent on specific tumor locations [24], e.g., soft tissue cannot be imaged using CT scans; ultrasound imaging cannot be performed on the lungs or bone; there are often drawbacks regarding spatial resolution; sometimes biomolecules are influenced within the body; there is the risk of over radiation exposure, low tissue penetration; PET and mMRI imaging is extremely expensive; and lastly, this technique is time consuming [24]. 

One extremely promising optical fluorescence modality for the diagnosis of MM is photodynamic diagnosis (PDD), also known as fluorescence diagnosis [25]. The few disadvantages of this modality can be avoided by low cost drug engineering and this diagnostic tool can be used in conjunction with Photodynamic Therapy (PDT) [26]. This combinative approach can allow for the early sensitive detection of MM, as well as simultaneously ensure a fast, non-invasive, reliable treatment for the eradication of MM cancer cells [27].

## 2. Photodynamic Diagnosis and Treatment

### 2.1. Photodynamic Diagnosis (PDD) and Photodynamic Therapy (PDT)

PDD is a diagnostic tool that has been recommended by the European Association of Urology (EAU) to detect carcinoma in situ within the bladder [28]. PDD involves the administration of a photosensitizer (PS), which is a fluorophore drug that is non-toxic to cells, as well as able to specifically localize in tumor cells only [25]. Low wavelengths of laser light, which range from 375 to 400 nm, is applied to the tumor, causing PS excitation [25]. Once the PS is excited, it fluoresces allowing for auto-fluorescent tumor identification [27]. This is an alternative form of diagnosis if applied and engineered correctly, it is accurate, rapid, and non-invasive and can be used to identify both primary and secondary tumors [25]. Moreover, since this diagnostic application utilizes a PS, PDT can be performed immediately after tumor identification, since the PS remains inactive after it fluoresces at the low wavelength and so can become reactivated at a higher wavelength to initiate immediate treatment [25,27]. 

PDT is an alternative treatment model that has been thoroughly investigated for various cancers [29]. This modality requires a light sensitive non-toxic PS drug to be administered to a patient [8]. Following specific uptake into localized tumor cells, it can become excited with light within the visible therapeutic window for PDT, which has a long wavelength range of 600 to 800 nm [29,30]. The excitation of the PS within this particular therapeutic wavelength range causes it to react with surrounding tissue molecular oxygen and so generates cytotoxic reactive oxygen species (ROS) and singlet oxygen (^1^O_2_), resulting in localized tumor destruction [8,29]. Overall, this PDT reaction encourages tumor destruction via various cell death pathways with the most prominent being autophagy and apoptosis [31]. 

Both PDD and PDT can be considered beneficial for MM diagnosis and treatment, as tumor sites are often located on or near the skin and so can be easily accessed with laser light, with minimal invasiveness. Secondly, due to the utilization of a PS, this photodynamic technique, which predominantly, as well as specifically (via molecular engineering) allows for PS drugs to specially localize in malignant tumor cells only and so allows for minimal side effects [29]. Lastly, combinative PDD and PDT can be considered effective, rapid, and sensitive, since these techniques allow for the simultaneous detection and treatment of cancer tumors [29,32].

#### 2.1.1. Photodynamic Modality Mechanism Action for PDD and PDT

The photodynamic modality of PDD and PDT is considered highly beneficial since it can be used for both detection and treatment of tumor cells [29]. This is due to its synergistic mechanism of action, whereby applying a PS drug and exciting it with various sources of laser light, in turn induces various biochemical excitations, which can cause either fluorescence for diagnosis or morphological responses such as cell death for treatment [29].

In general, PDD is a diagnostic tool which utilizes a PS, that becomes excited when a light source, which is applied within the short purple to blue wavelength ranging from 330 to 400 nm [25,33,34]. In principle, this PDD short wavelength application causes a tumor localized PS to fluoresce, inducing a desirable optical effect, which allows for specific identification of tumor cells in the body, while not causing cellular damage to the tumor or normal body cells [33].

PDT is a cancer treatment modality, which also requires the administration of a PS drug, which becomes localized in tumor cells [35]. However, the difference between PDD and PDT is that within PDT applications the PS is excited using a longer therapeutic window wavelength of light that ranges from 600 to 800 nm [35]. When this particular light source range excites a PS drug, a photo-oxidative reaction occurs, whereby the excited PS reacts with molecular tissue oxygen to produce hydroxyl radicals and/or superoxide anions, which are collectively cytotoxic ROS and ^1^O_2_ [35]. These cytotoxic species induce oxidative stress within tumor cancer cells initiating cell death pathways, which ultimately cause tumor destruction [35].

Since PDD and PDT have the same requirements, such as a PS drug that can localize within a tumor, with the only difference being the different wavelengths of light being applied to cause PS excitation for either diagnostic auto-fluorescence or cytotoxic species induction for treatment, this photodynamic modality can be applied simultaneously (Figure 1) [33]. 

The mechanism of action for PDD is as follows: once the PS becomes localized within tumor cells and a short wavelength of light is applied, it interacts with the PS causing it to change from a ground single state to an excited single state, allowing it to fluoresce for diagnosis [34]. The PDT mechanism of action differs from PDD in that once the PS becomes localized within tumor cells, a high wavelength of light is applied, causing ground single state PS to become excited to single state [32]. Then, via intersystem crossing, this excited PS singlet state becomes excited to a triplet state, which reacts with surrounding molecular oxygen to create either type I or type II cytotoxic oxygen species [34]. Both PDT reaction type I and type II are oxygen-dependent, however their end cytotoxic species production is highly dependent on the oxygen concentration within a tumor cell [8]. Within type I reaction mechanisms, the oxygen concentration within a tumor cell is low and so superoxide ROS is generally produced, whereas within type II reaction mechanisms the tumor oxygen concentration tends to be higher and so ^1^O_2_ is formed [34]. Within both reaction types cytotoxic species are produced, which in turn cause the tumor cell to undergo oxidative stress and so are destroyed via various apoptotic and autophagy cell death pathways [31]. However, type II reactions are generally more favorable forms of cellular tumor destruction as ^1^O_2_ is extremely reactive and so easily damages tumor cells, with a more favorable PDT cytotoxic outcome [35].

#### 2.1.2. Specific Challenges and Limitations of PDD and PDT in Relation to MM

The challenges faced when applying PDD and PDT for cancer diagnosis and treatment are often due to the shortcomings of the PS drug. 

Firstly, the PS drug must have two significant absorbance values within the short and high wavelength light ranges in order to allow for combinative excitation for diagnosis, as well as treatment [33]. Therefore, in order to ensure the effectiveness of the photodynamic method when applying different wavelengths of light, the correct exposure time and wavelength needs to be applied in order to excite the PS. Moreover, the different applied wavelengths of light need to allow for the excited PS to be able to fluoresce for PDD, as well as to allow the PS to be able to produce cytotoxic oxygen species for PDT tumor destruction [36]. 

In relation to the tumor site, the accessibility of the tumor with the laser can sometimes affect the overall success of PDD and PDT. Therefore, deep-seated tumors that are inaccessible to light penetration need alternate procedures to be investigated [37]. Additionally, MM tumors, which are highly pigmented with melanin or very thick in appearance, sometimes do not permit for efficient low laser wavelengths of light to reach their core and so PS excitation for diagnosis can become difficult. However, in most cases, high laser wavelengths of light can’t penetrate them for effective PDT [38,39]. Thus, for combinative PDD and PDT treatments of MM tumors, extensive investigation needs to be performed as to ensure the PS drugs of choice is optimized to allow for deep-seated tumor light penetration and overall identification and diagnosis [38,39]. 

Lastly, the PS concentration, in relation to specific uptake and localization within tumor cells is imperative, as to ensure that there is efficient fluorescence for diagnosis, as well as high ROS and ^1^O_2_ yields for effective cell death, when excited at various wavelengths [40]. Thus, there has been a huge drive in current focused research to develop multicomponent biomarker functionalized PS drugs combined with nanocarriers in order to improve PS drug selective and targeted uptake in tumor cells, so its overall concentration is increased in order to enhance this combinative photodynamic cancer diagnosis and treatment and modality [39].

### 2.2. Photosensitizers (PSs) for MM PDD and PDT Applications 

PSs are light sensitive non-toxic dyes that absorb light at appropriate wavelengths to either cause fluorescence, phosphorescence, or cellular damage [41]. In order for a PS to be considered ideal within combinative PDD and PDT applications, they need be able to localize within the tumor cells as well as have enhanced and selective cellular uptake [33,42,43]. Additionally, the PS of choice need to be able fluoresce at low wavelengths of light, while at higher wavelengths of light be able to generate enough cytotoxic species to induce cell death [33,43]. Moreover, PSs must be chemically pure, stable, and hydrophobic; allow for chemical modification; be amphilic in configuration; have minimal dark toxicity within cells; and be able to be rapidly removed from the body [42,43]. Since the early 1990s, many natural and synthetic PS dyes have been tested and applied within PDT research experiments. However, little focus has been done in relation to investigating PS for combinative photodynamic cancer diagnosis and treatment [41]. Generally, PSs are classified based on their functionality and so are placed into three groupings, i.e., first generation, second generation, and third generation [29]. Additionally, PS are divided into four main prominent classes based on their structure, i.e., porphycenes, chlorins, porphyrins, and phthalocyanines [43]. 

#### 2.2.1. First-Generation PSs for Simultaneous MM PDD and PDT Applications

First-generation PS have been extensively investigated in PDT applications [31]. Porphyrins are first generation PSs, are stable compounds, have extremely poor tissue penetration depth, and induce unwanted photosensitivity [8]. 

The first PS that was approved for use in PDT and PDD investigations was the purified hematoporphyrin derivative Photofrin; it noted operational killing of melanoma, breast, bladder, lung, oesophageal, gastric, ovarian, and cervical cancer [42,43]. However, Photofrin noted numerous disadvantages in PDD and PDT applications; it was retained by cutaneous tissue resulting in severe photosensitivity over a prolonged periods of time, and was ineffective in producing high singlet oxygen yields at 630 nm (even though it fluoresced at lower wavelengths for diagnosis), which caused poor tissue penetration depth and so overall minimal cell death [8,42,43]. 

5-Aminolevulinic acid (5-ALA) and 5-methylaminolevulinic acid (5-MAL) are enzymatically converted to protoporphyin IX (PpIX) is converted to a heme compound containing an iron ion bound to a porphyrin with/without axial ligands [42]. 5-ALA is currently the only first-generation PS that has been widely utilized in dermatology, since it has shown promising results for PDD, due to its ability to fluoresce when excited within a wavelength range of 375 to 445 nm (visible blue light), as well as has been noted to have superficial MM PDT treatment effectiveness at 630 nm (visible red light) [44]. However, overall the numerous limitations of first-generation PSs have created the need to develop and further research PSs, which have superior properties allowing them to be considered ideal drugs of choice for combinative PDD and PDT applications.

#### 2.2.2. Second-Generation PSs for Simultaneous MM PDD and PDT Applications

Second-generation PSs were investigated and developed in order to overcome some of the limiting setbacks first-generation PSs noted [8]. Benzoporphyrins, chlorins, hypericin, and phthalocyanines are examples of second-generation PSs that have been investigated for the use in MM PDT treatment applications and some are currently in advanced stages of PDT clinical trials [45,46]. It has been noted that chlorins have a high PDT efficacy rate when treating basal cell carcinomas and squamous cell carcinomas [8].

Benzoporphyrins are also known as Verteporfin. They are hydrophobic PSs that are effectively activated at long 690 nm wavelengths for effective deep tissue MM penetration and overall PDT treatment, as well as allowing for fast and effective removal/clearance. However, Verteporfin cannot be activated at low light wavelengths for PDD [42]. Some second-generation porphyrins PSs that have been successfully applied in MM PDT treatment include, i.e., Verteporfin, 10,15,20-tritolylporphyrin-5-(4-amidophenyl)-[5-(4-phenyl)-10,15,20-tritolyporphyrin] (T-D), Halogenated porphyrins, 5,10,15,20-tetrakis (2,6-difluoro-3-*N*-methylsulfamoylphenyl) bacteriochlorin, Meso-tetrakis-(4-sulfonato phenyl) porphyrin (TPPS_4_), and Ruthenium porphyrins [8].

Chlorins are derived from porphyrins and they are tetrapyrrole molecules that contain a pyrrole ring that has at least one less double bond present [47]. This alteration to a Chlorins structure shifts its absorption band to 650–690 nm, allowing it to have a strong MM PDT application in the far-red region. However, no MM PDD application has been investigated [47]. Bacteriochlorins are derivatives of chlorin and have a second double bond removed from their second pyrrole ring, which shifts their absorption band further into the far-red region for a more MM effective PDT treatment. However, they have no MM PDD abilities [47]. Furthermore, the overall stability of bacteriochlorin has been questioned, as has the common occurrence of photobleaching [47]. Thus, these types of Chlorin PS really only have successful application in MM PDT treatment and not for MM PDD. The only Chlorin derivative that has been studied for photodynamic applications in head, neck, and MM cancer patients is Talaporfin Sodium (also known as Laserphyrin), since it achieves PDD fluorescence at 408 nm, with effective PDT cell death outcomes. However, severe and prolonged photosensitivity of two weeks subminimum has rendered it non-idealistic [42].

Hypericin is a natural pigment from the *Hypercium perforatum* plant. It is a non-porphyrin-based PS, which is known to have selective passive tumor uptake properties, with minimal dark toxicity and photobleaching, when compared to another porphyrin. However, it has been found to be unstable in solution, as well as to exhibit a low solubility [45,48]. Within the photodynamic treatment of squamous cell carcinoma, hypericin was used as a PDD fluorescent marker within the 320 to 400 nm light wavelength range, as well as used for PDT treatment since it has a strong absorption peak from 543 to 590 nm, with effective outcomes [45,49]. Other studies have noted the effective treatment of MM with hypercin-mediated PDT [50]; however, no investigations into its combinative MM PDD abilities have been published. 

Phthalocyanines are extremely effective PSs; they are porphyrin-based PSs that contain a central metal atom consisting of either a zinc, silicon, or aluminum. This central metal atom allows for idealistic PS properties [32]. Since, phthalocyanines display strong red-light absorption at 675 nm and have strong secondary absorption within the 330 to 400 nm blue light range, they can be retrospectively and successfully applied in combination within both MM PDT and PDD applications [45]. Moreover, phthalocyanines exhibit idealistic properties of rapid clearance from the body, reduced photosensitivity, and high fluorescent signal generation at low light wavelengths and high ^1^O_2_ generation (due to their central metal ion) at high light wavelengths, allowing for combinative photodynamic applications to be effective [42,45]. Additionally, phthalocyanine PSs can be further modified with either cationic or sulfonic acid groups to improve their overall water solubility or liposomal encapsulation, as well as to allow for further functionalization of drug delivery via nanocarrier conjugation [32,42]. Sulfonated aluminum and silicone phthalocyanines have shown promising results within clinical and stage II trials for various cancers, due to their minimal dark toxicity and fluorescence. However, no studies have researched their application within photodynamic MM diagnosis and treatment [42]. The PSs AlPcS_4_Cl (Aluminum (III) phthalocyanine chloride tetra-sulphonate) and ZnODPc (zinc (II)-octadecylphthalocyaninato) have been noted to successfully localize in MM cells, with high passive tumor selective absorption properties and abilities to induce significant PDT cell death at high wavelengths of light. However. their PDD abilities at short wavelengths of light within MM remains ongoing [45].

#### 2.2.3. Third-Generation PSs for Simultaneous MM PDD and PDT Applications

Third-generation PSs are first-generation or second-generation PSs that are further functionalized with various targeting strategies. This further PS functionalization can include either bio-targeting antibodies or ligands to identify over expressed markers in tumor cells for specifically enhanced active PS uptake recognition or carriers for enhanced PS passive absorption. In some cases, third-generation PS are modified with both biomarkers and carriers for improved overall PS subcellular uptake in tumor cells for enhanced concentration and so improved PDD and PDT outcomes [51]. Carriers mostly include nanoparticle (NP) systems and liposomes, whereas targeting agents include antibodies, aptamers, and peptides [8,31]. Thus, third-generation PSs drugs can be divided based on their targeting abilities, i.e., an active targeting strategy makes use of a PS conjugated biomarker antibody, aptamer, or peptide system for specific molecular recognition uptake. A passive targeting strategy, however, makes use of NP carrier system, which mimics biological molecules and so avoids immune system barriers for enhance uptake [8]. 

### 2.3. Nanoparticles and Biomarkers for Enhanced PSs MM Uptake

Nanoparticles (NPs) have been investigated extensively in order to promote PS drug delivery in PDT [32,52]. NPs exhibit ideal features that make them appropriate for PS drug uptake enhancements, since they can become transport carries to promote passive tumor uptake due to the enhanced permeability and retention (EPR) effect [8]. NPs that are used for drug delivery systems can exploit flaws with regards to tumor microvasculature allowing for drug accumulation in tumors through the EPR effect [53]. However, it has been noted that the EPR effect is not an effective moiety, since there are additional factors requiring consideration such as heterogeneous tumor vasculature and high interstitial fluid pressure [53]. Furthermore, PS conjugated NPs promote passive tumor absorption due to their small dimensions, allowing for easy accumulation within cancer cells and due to a NPs large surface area to volume ratio they can support a large loading quantity of PS therapeutic agents, allowing for a more concentrated drug delivery [54]. Additionally, NPs have ability to be undetected by the immune system, since they mimic biological molecules and so protect PS from immunological destruction [32]. Lastly, NP functionalization with PS drugs can improve overall stability, solubility and permeability of the final drug delivery system [31,32]. Examples of NP drug delivery systems which have been investigated for enhanced PS passive drug uptake and improved PDT outcomes, include polymeric particles, liposomes and micelles, ceramic, dendrimers, silica, alumina organic, and metal oxide-based NPs [8]. To date, the only NPs that passively improved PS drug absorption systems for improved MM PDT outcomes, have been when using silica, gold, magnetic, or chitosan NPs [8]. Within PDD cancer applications 5-ALA, hypericin, and phthalocyanines have been successfully conjugated to NPs (such as polymeric, solid-lipid and magnetic). However, none of these PS nanocarrier systems have been investigated for the successful diagnosis of MM [55].

Within active drug delivery strategies, a PS drug is transported to a specific target tumor site via a molecular recognition process [8]. Targeting biomolecules that are exploited in active targeting PS drug delivery systems for tumor cells include; monoclonal antibodies (mAb), aptamers, antibody fragments, peptides, and/or DNA/RNA [8]. Thus, targeting biomolecules are either bound to PS drugs alone or conjugated onto PS NP carrier systems in order to promote a more targeted active uptake of PS [56]. When PSs are functionalized with active targeting biomolecules that specifically bind to receptors, which are only over expressed by tumor cells nuclear receptor sites, malignant cell membranes or cytoplasmic receptor sites, the delivery of the PS becomes enhanced and so concentrated, as the active drug molecular recognition system will only allow for PS accumulation within targeted tumor regions [32]. Examples of NP drug delivery platforms that actively enhance PS drug targeting in PDT are generally inorganic nanomaterials such as: quantum dots, solid lipids, self-illuminating nanocrystals, theranostic, hydrogels, immunoconjugates, metal-oxide based, or upconverted NPs [57]. Thus, biomarker PS drug delivery allows for only specifically enhanced compatible target tumor cell drug delivery and so healthy cells remain undetected or unharmed within photodynamic cancer diagnosis or treatment regimes. Various studies have reported that MM tumor cells overexpress various biomarker cell surface receptors, such as TRAIL-receptor, integrin anß3, a combination of Drosophilia protein and Caenorhabditis elegans protein, extracellular matrix 1, B-cell lymphoma 2, mitochondrial p32 protein, integrin alpha 4 beta 1 protein, ephrin type-A receptor 2, and melanoma inhibitory activity (MIA) [58,59]. Additionally, rituximab, bevacizumab, and trastuzumab are FDA approved monoclonal which can be used as biomarkers to target MM cells [8,60]. However, to date, no reports have been made whereby successful third-generation PS drugs have been developed that have active biomarker targeting strategies and/or NP passive uptake abilities for enhanced PS uptake for improved PDT and PDD abilities in MM.

Chemophototherapy is an emerging combinative treatment modality that utilizes chemotherapeutic drugs such as Doxorubicin or Cisplatin and NP enhanced PDT [53]. This is an effective treatment as both treatment modalities cause damage to the cancer cells [53]. The NPs utilized in this modality are effective due to increased drug uptake caused by the photo-induced enhanced vasculature permeability effect [53]. NPs that have been utilized include but are not limited to: spiropyran-based NPs, self-assembled lipid-polymer hybrid NPs, nanographene oxide NPs, and polymeric micelles [53]. This has the added advantage of bioavailability and improved efficacy due to favorable anti-tumor responses [53]. The disadvantages of this treatment modality are photosensitivity, penetration depth, immunosuppression, and side effects related to the chemotherapeutic agents [60]. Balancing the dosimetry of the PS drug with the chemotherapeutic agent and the NP is very difficult. Thus, even though this is a promising strategy, further investigation in relation to MM is warranted [53].

## 3. Conclusions

PDT cannot be used to treat aggressive systemic MM; it can only be used to treat localized MM. Currently, surgery is the established and effective care for localized MM. However, such surgery is invasive and leaves unwanted scaring. Other standard treatments for MM include radiation therapy, biological therapy, and chemotherapy, with immunotherapy and targeted therapy recently emerging. However, alternate modalities, such as PDT, are non-intrusive and not affected by resistance factors and immune-related side effects. Further, as well as can be used in combination for treatment and diagnosis and so definitely should be considered a far better treatment modality. From this review article, it can be concluded that most photodynamic MM research is centered around PSs for PDT applications. PDD has been thoroughly investigated as an early diagnostic tool for bladder cancer [28]. However, for other cancers such as MM, it still remains under-investigated. The first-generation PS 5-Ala is the only PS that has been thoroughly investigated for PDD application, but it is not specific to melanoma. Since PSs such as 5-ALA, Hypericin, and Phthalocyanines have been reported to note significant MM PDT treatment, they should be further investigated for MM PDD, if they can fluoresce at lower wavelengths.

Additionally, despite significant efforts to develop PS drugs that are modified with either active targeting biomolecules or passive NPs, which will enhance tumor uptake and improve outcomes of combinative PDD and PDT, to date no published contributions towards investigating this photodynamic modality for MM has been made. Therefore, it requires extensive investigation [61]. Overall, the contribution of research to future smart PS drug systems that allows for the photodynamic diagnosis and treatment of MM should improve early diagnosis and treatment prognosis outcomes of patients. Moreover, this would allow for the development of an extremely sensitive, fast, cost effective, and non-invasive combinative diagnostic and treatment tool for MM. Therefore, simultaneous PDD and PDT treatment of MM in when applying 5-ALA, Hypericin, and Phthalocyanines PS drugs requires ongoing investigation to allow these two photodynamic methods to synchronize with one another.

## Figures and Tables

**Figure 1 molecules-24-03153-f001:**
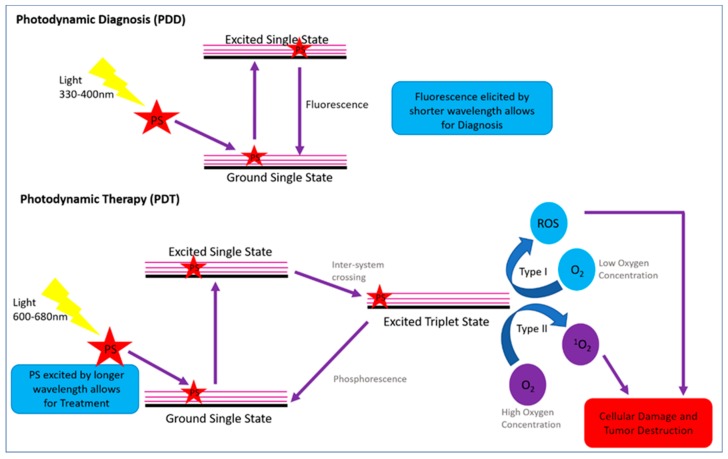
Mechanism of PDD Action verses PDT Action.

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
