# Peer review of "Simultaneous Photodiagnosis and Photodynamic Treatment of Metastatic Melanoma"

_molecules, 2019, doi:10.3390/molecules24173153_

Round 1
Reviewer 1 Report
The paper entitled „Combinative Photosensitizer Drug Delivery Systems for the Simultaneous Photodiagnosis and Photodynamic Treatment of Metastatic Melanoma” by Channay Naidoo, Cherie Ann Kruger, and Heidi Abrahamse provide an overview theranostic photosensitizer delivery tools. The chapter “Introduction” is extremely short and do not lead readers to reviewed the problem. Further chapters describing melanoma properties, and its conventional diagnosis should be shortened and included in “Introduction”. Chapter 3.1 and 3.1.1 can be shortened and included in the “Introduction”. Moreover, this information is well-known and reviewed many times before. Figures 1, 2, 3, 4 are unnecessary well-known schemes and structures. In the chapter 3.1.2 Authors have presented specific limitations of the PDD and PDT for diagnosis and treatment of MM as well, whereas the title “Challenges and Limitations of PDD and PDT” suggest only general issues. This can be confusing for the readers. In chapters 3.2; 3.2.1 and 3.2.2 Authors have described first and second-generation photosensitizers in well-known manner published before. In these chapters, the information concerning melanoma metastasis is curt. The reviewed topic is described and is not discussed in the few lines of the chapter 3.2.3.1. Chapter 4 is missing. In chapter 5 “Conclusions” there is lack of conclusion. In the overall assessment, the articles can not to be published. I recommend rejecting.
Author Response
Please see attatchment

Reviewer 2 Report
In this work, the authors present "Combinative Photosensitizer Drug Delivery Systems for the Simultaneous Photodiagnosis and Photodynamic Treatment of Metastatic Melanoma".
Overall, the review reads very well and presents useful information, especially about the background of melanoma. However there are some concerns that should be addressed:
1) The first part of the title of the manuscript is "Combinative Photosensitizer Drug Delivery Systems". The use of the words combinative and drug delivery implies this refers to chemotherapy and PDT. However, in this sense, little material - virtually none at all - is presented on that particular topic. Therefore, I recommend removal of the first 7 words of the title.
2) BCC and HCC are not treated in the same way as melanoma, so the authors need not mention "1 in every 3 cancer patients are diagnosed with skin cancer." in the abstract.
3) The authors do a good job in describing some current melanoma practices. However, the authors should mention what are the current treatments for metastatic melanoma (including checkpoing blockade, BRAF drugs, which chemo drugs like taxanes,etc ),
4) Authors should discuss what the rationale there would be for a local treatment (PDT) of an aggressive systemic disease. Would it be for palliative purpose only, since presumably it would not impact survival? If the disease is localized, surgery would likely remain the standard of care, as it is well-established and effective
5) In the nano-section, the authors should mention that nanoagents are able to integrate PDT with chemotherapy and this area has been the focus of emerging interest (Luo et al. Advanced Science, 4, 2017, 1600106)
Round 2
Reviewer 1 Report
The paper entitled „Combinative Photosensitizer Drug Delivery Systems for the Simultaneous Photodiagnosis and Photodynamic Treatment of Metastatic Melanoma” by Channay Naidoo, Cherie Ann Kruger, and Heidi Abrahamse provide an overview theranostic photosensitizer delivery tools. After provided improvements the article can be published.